# Evaluating fairness of machine learning prediction of prolonged wait times in Emergency Department with Interpretable eXtreme gradient boosting

**Hao Wang[ID]1\*, Nethra Sambamoorthi2, Nathan Hoot1, David Bryant1, Usha Sambamoorthi3**

**1** Department of Emergency Medicine, JPS Health Network, Fort Worth, Texas, United States of America, **2** Senior biostatistician, CRM Portals LLC, Fort Worth, Texas, United States of America, **3** College of Pharmacy, University of North Texas Health Science Center, Fort Worth, Texas, United States of America

\* hwang@ies.healthcare

## Abstract

It is essential to evaluate performance and assess quality before applying artificial intelligence (AI) and machine learning (ML) models to clinical practice. This study utilized ML to predict patient wait times in the Emergency Department (ED), determine model performance accuracies, and conduct fairness evaluations to further assess ethnic disparities in using ML for wait time prediction among different patient populations in the ED. This retrospective observational study included adult patients (age ≥18 years) in the ED (n=173,856 visits) who were assigned an Emergency Severity Index (ESI) level of 3 at triage. Prolonged wait time was defined as waiting time ≥30 minutes. We employed extreme gradient boosting (XGBoost) for predicting prolonged wait times. Model performance was assessed with accuracy, recall, precision, F1 score, and false negative rate (FNR). To perform the global and local interpretation of feature importance, we utilized Shapley additive explanations (SHAP) to interpret the output from the XGBoost model. Fairness in ML models were evaluated across sensitive attributes (sex, race and ethnicity, and insurance status) at both subgroup and individual levels. We found that nearly half (48.43%, 84,195) of ED patient visits demonstrated prolonged ED wait times. XGBoost model exhibited moderate accuracy performance (AUROC=0.81). When fairness was evaluated with FNRs, unfairness existed across different sensitive attributes (male vs. female, Hispanic vs. Non-Hispanic White, and patients with insurances vs. without insurance). The predicted FNRs were lower among females, Hispanics, and patients without insurance compared to their counterparts. Therefore, XGBoost model demonstrated acceptable performance in predicting prolonged wait times in ED visits. However, disparities arise in predicting patients with different sex, race and ethnicity, and insurance status. To enhance the utility of ML model predictions in clinical practice, conducting performance assessments and fairness evaluations are crucial.

**Data availability statement:** We provided our deidentified data in a Supplemental file.

**Funding:** The project described was supported by the National Institute on Minority Health and Health Disparities through the Texas Center for Health Disparities (NIMHD) 5S21MD012472-05 (US), and the National Institute of Health/Artificial Intelligence/Machine Learning Consortium to Advance Health Equity and Researcher Diversity Grant # 1OT2OD032581-01 (US). The content is solely the responsibility of the authors and does not necessarily represent the official views of the NIH.

**Competing interests:** The author has declared that no competing interests exist.

## Author summary

We used machine learning to predict patient wait times in the Emergency Department (ED), evaluated the model's performance, and conducted fairness assessments. We did a retrospective study including adult patients (age ≥18 years) in the ED (n=173,856 visits with 99,178 unique patients) who were assigned an Emergency Severity Index (ESI) level of 3 at triage. We defined prolonged wait time as a waiting time of ≥30 minutes, and we employed extreme gradient boosting (XGBoost) to predict these ED visits with prolonged wait times. Nearly half of the ED visits in our study experienced prolonged wait times. Our XGBoost model demonstrated moderate predictive accuracy. However, we observed disparities in the model's performance across different demographic groups, including sex, race and ethnicity, and health insurance. To ensure the practical applicability of artificial intelligence and machine learning models in clinical settings, it is essential to perform both accuracy assessments and fairness evaluations.

## Introduction

Healthcare quality can be influenced by various factors, including patient demographics, languages and cultures, psychosocial factors, patients' perceptions, patient-provider communication, and conditions within healthcare facilities.[1–3] Specifically, prolonged waiting times in the Emergency Department (ED) can diminish patient satisfaction, increase rates of incomplete care (e.g., patients leaving without being seen), and worsen patient clinical outcomes. [4–6] Accurate prediction of patient wait times in the ED holds promise for alleviating patient anxiety, improving patient flow management, and enhancing overall care quality.

In recent years, there has been growing reports for the integration of artificial intelligence (AI) and machine learning (ML) in the field of medicine, including applications ranging from diagnostics to treatment assistance.[7,8] For instance, by leveraging pattern recognition and association learning algorithms, ML models can achieve high accuracy in identifying individuals at risk for Alzheimer's disease, thereby enhancing disease diagnosis in clinical practice. [9] ML algorithms can also identify demographic factors, such as age, sex, and poverty ratio, that are strongly associated with COVID-19-related mortality, providing valuable insights for disease management.[10] Additionally, numerous ML models have been derived to forecast patient wait times in the ED, demonstrating acceptable levels of performance accuracy.[11,12] However, while these models offer overall accuracy metrics, their ability to predict outcomes at the individual level may be limited. Evidence suggests that systematic biases may emerge during the development of ML models. While these models may exhibit fairness in their overall performance accuracy, they can still yield unfair outcomes for certain individuals.[13] Common sources of unfairness include patient demographics: such as sex, race and ethnicity, or socioeconomic status (SES), often known as sensitive attributes.[14,15] Consequently, these models may perform better for some patient groups while delivering suboptimal results for others.

To prevent systematic biases that could undermine the accuracy of model predictions, conducting a thorough fairness evaluation of ML models is essential. This evaluation should include both group-level and individual-level assessments to detect any biases. Upon identifying unfairness, appropriate adjustments made to the ML model may enhance the accuracy of the predictions and fairness in clinical applications. Only through such rigorous assessments can ML models be deemed reliable for providing accurate predictions. Given the rapid advancement of ML healthcare models in Emergency Medicine, these evaluations offer critical

insights for establishing standards and implementing quality control measures for the use of ML in healthcare.

In this study, our objectives are: 1) to predict prolonged ED wait time using ML, and 2) to evaluate fairness predictions across sensitive attributes at both group and individual levels. Fairness evaluations of ML predictions are critical for health equity relevant quality assessment to reduce healthcare disparities.

## Results

### Descriptive analysis

A total of 173,856 adult patient visits were included in this study. Sensitive attributes consistent of male versus female, Non-Hispanic White (NHW) versus Non-Hispanic Black (NHB) versus Hispanic, and having health insurance versus no health insurance. We observed that more ED visits from females (52.63%), Hispanics (54.00%), and patients without health insurance (52.34%) tended to have a prolonged wait times than those from males (43.93%), NHW/NHB individuals (43.68%/46.66%), and patients with health insurance (46.66%), as illustrated in Fig 1. General characteristics of the study patients are listed in Table 1. It was also found that a greater proportion of visits from younger patients non-English speakers, who arrived not via ambulance, who presented at triage with normal vital signs exhibited prolonged wait times compared to their counterparts. Additionally, more patients who presented at the ED under overly-crowded conditions tended to wait ≥30 minutes than those presented under the ED not-crowded conditions (p<0.001, Table 1).

### Main results

**Machine learning algorithms can be used to predict patient wait times.** In this study, XGBoost was developed to predict ED visits with prolonged wait time. Table 2 shows the overall performance accuracies when XGBoost was used for prolonged wait time prediction. In addition, Table 2 also presents the fairness evaluation across different sensitive attributes (i.e., XGBoost model was trained on the entire dataset and then applied to different sensitive attributes). Specifically, concerning sex differences, the XGBoost model demonstrated

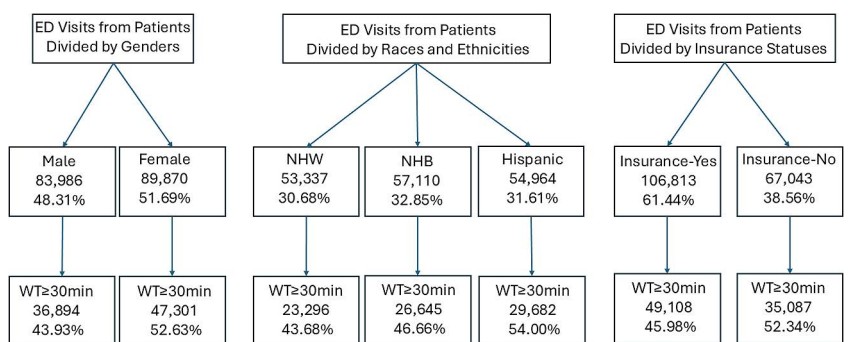

**Fig 1. ED Visits from Patients of Prolonged Wait Time by Sex, Race and ethnicity, and Insurance coverage.** Fig 1 shows a total of 173,856 adult ED visits from patients with three subgroups (sex, race and ethnicity, and insurance). ED visits refer to a total of 173,856 adult patient visits assigned with emergency severity index -3 (ESI) at triage upon arrival at the ED. The percentages of ED visits from patients with prolonged wait times (i.e., Wait Times≥30min) in each subgroup are shown. It was found that more ED visits from female patients, Hispanic patients, and patients without insurance had prolonged wait times than ED visits from male patients, NHW/NHB patients, and patients with insurance. Abbreviations: NHW, Non-Hispanic White; NHB, Non-Hispanic Black.

**Table 1. Characteristics of the study ED Visits.**

**Comparisons of all variables by Wait-time among Emergency Department Visits Triaged as ESI-3 Level**

| | Wait Time<30min | Wait time≥30min | P value |
|---|---|---|---|
| Number of patient visits | 89,661 (51.57) | 84,195 (48.43) | |
| Age --- year Median (IQR) Mean (SD) | 45 [32-58] 46 (16) | 43 [31-55] 43 (15) | <0.001 <0.001 |
| Sex --- n (%) Male Female | 47,092 (56.07) 42,569 (47.37) | 36,894 (43.93) 47,301 (52.63) | <0.001 |
| Marital status --- n (%) Single Married Others | 48,195 (53.04) 21,867 (47.01) 19,599 (53.73) | 42,668 (46.96) 24,649 (52.99) 16,878 (46.27) | <0.001 |
| Race and ethnicity --- n (%) NHW NHB Hispanic/Latino NHA Others | 30,041 (56.32) 30,465 (53.34) 25,282 (46.00) 1,220 (45.14) 2,653 (46.20) | 23,296 (43.68) 26,645 (46.66) 29,682 (54.00) 1,483 (54.86) 3,089 (53.80) | <0.001 |
| Language --- n (%) English Spanish Others | 77,177 (53.11) 10,008 (43.97) 2,476 (42.88) | 68,145 (46.89) 12,752 (56.03) 3,298 (57.12) | <0.001 |
| Insurance --- n (%) No Yes | 31,956 (47.66) 57,705 (53.34) | 35,087 (52.34) 49,108 (46.66) | <0.001 |
| Primary care physician --- n (%) Assigned Not assigned | 34,541 (48.98) 55,120 (61.48) | 35,976 (51.02) 48,219 (57.27) | <0.001 |
| Comorbid --- n (%) No One Multimorbidity | 38,019 (50.55) 14,647 (50.01) 36,995 (53.35) | 37,196 (49.45) 14,644 (49.99) 32,355 (46.65) | <0.001 |
| Crowding status --- n (%) Not-crowded Crowded Overly-crowded | 32,540 (79.29) 35,239 (52.85) 21,882 (33.08) | 8,500 (20.71) 31,437 (47.15) 44,258 (66.92) | <0.001 |
| Mode of arrival --- n (%) Private car Ambulance Public transportation Ambulatory | 40,633 (39.12) 40,898 (82.86) 489 (36.14) 7,641 (39.66) | 63,245 (60.88) 8,460 (17.14) 864 (63.86) 11,626 (60.34) | <0.001 |
| Clinical hours --- n (%) Within clinical hour Out of clinical hour | 44,042 (51.18) 45,619 (51.95) | 42,006 (48.82) 42,189 (48.05) | 0.001 |
| Weekday vs. weekend --- n (%) Weekday Weekend | 62,296 (48.49) 27,365 (60.29) | 66,171 (51.51) 18,024 (39.71) | <0.001 |
| Having High BP at Triage --- n (%) Yes No | 51,150 (50.81) 38,511 (52.62) | 49,523 (49.19) 34,672 (47.38) | <0.001 |
| Having abnormal vital signs at triage (exclude high BP) --- n (%) Yes No | 21,849 (57.20) 67,812 (49.99) | 16,347 (42.80) 67,848 (50.01) | <0.001 |

*(Continued)*

**Table 1.** (Continued)

Note: Based on 173,856 ED visits of two groups (wait time<30min and wait time≥30min) from January 1, 2019, to December 31, 2021. Categorical variables were compared using the Chi-square test. Continuous variables were compared either using Student-t test (mean) or using Kruskal-Wallis' test (median). p<0.001 among all variables when two groups were compared. Abbreviations: ESI, Emergency Severity Index; n, number; NHW, Non-Hispanic White; NHB, Non-Hispanic Black; NHA, Non-Hispanic Asian; IQR, Interquartile range; SD, Standard deviation; BP, Blood pressure.

**Table 2.** XGBoost model performance accuracy, fairness evaluation, and counterfactual analysis across different sensitive attributes.

| | Accuracy | Precision | F1 score | AUROC | Recall (TPR) | Specificity (TNR) | FPR | FNR |
|---|---|---|---|---|---|---|---|---|
| **Overall** | **0.75** | **0.72** | **0.75** | **0.81** | **0.80** | **0.71** | **0.29** | **0.21** |
| Fairness evaluation across different sensitive attributes | | | | | | | | |
| Sex | | | | | | | | |
| Male | 0.75 | 0.69 | 0.73 | 0.81 | 0.76 | 0.74 | 0.26 | 0.24 |
| Female | 0.75 | 0.74 | 0.77 | 0.81 | 0.82 | 0.67 | 0.33 | 0.18 |
| Race and ethnicity | | | | | | | | |
| NHW | 0.75 | 0.70 | 0.73 | 0.81 | 0.76 | 0.75 | 0.25 | 0.24 |
| NHB | 0.75 | 0.72 | 0.74 | 0.81 | 0.76 | 0.74 | 0.26 | 0.24 |
| Hispanic | 0.75 | 0.73 | 0.78 | 0.81 | 0.85 | 0.62 | 0.38 | 0.15 |
| Insurance | | | | | | | | |
| No-Insurance | 0.74 | 0.72 | 0.77 | 0.80 | 0.83 | 0.65 | 0.35 | 0.17 |
| Yes-Insurance | 0.75 | 0.71 | 0.74 | 0.82 | 0.77 | 0.74 | 0.26 | 0.23 |
| Counterfactual analysis across different sensitive attributes | | | | | | | | |
| Sex | | | | | | | | |
| Female → Male | 0.75 | 0.72 | 0.75 | 0.81 | 0.79 | 0.71 | 0.29 | 0.21 |
| Race and ethnicity | | | | | | | | |
| NHB → NHW | 0.75 | 0.71 | 0.75 | 0.81 | 0.79 | 0.71 | 0.29 | 0.21 |
| Hispanic → NHW | 0.75 | 0.72 | 0.75 | 0.81 | 0.79 | 0.71 | 0.29 | 0.21 |
| Insurance | | | | | | | | |
| No- → Yes-Insurance | 0.75 | 0.72 | 0.75 | 0.81 | 0.79 | 0.71 | 0.29 | 0.21 |

Note: XGBoost was used for prolonged wait time predictions. When the overall model was used to predict wait time on different sensitive attributes (sex, race and ethnicity, and insurance coverage), disparities occurred. When used for clinical application focusing on FNR, this model favored to predict prolonged wait time among ED visits from female patients, Hispanic patients, and patients without insurance indicating the occurrence of ethic disparity from ML prediction model. In terms of the counterfactual analyses, keeping all other features unchanged, the performance accuracies were reported when female was changed to male, NHB was changed to NHW, Hispanic was changed to NHW, and patients without insurance were changed to patients having insurance. Abbreviations: XGBoost, eXtreme Gradient Boosting; NHW, Non-Hispanic White; NHB, Non-Hispanic Black; AUROC, area under the receiver operating characteristic curve; FNR, False Negative Rate.

different accuracies in predicting prolonged wait times among male and female individuals. The model predicted higher FPR (33% vs. 26%) and lower FNR (18% vs. 24%) in females when compared to male individuals. However, in clinical practice, greater emphasis is placed on minimizing the FNR, indicating cases where patients experienced prolonged wait times despite the model predicting otherwise. Utilizing the XGBoost model, the FNR was 18% when predicting prolonged wait times among female patients, whereas it was 24% when the same model was applied to male patients (Table 2). Disparities were also observed among patients of different races and ethnicities, as well as among those with or without insurance coverage (Table 2). Overall, in terms of the clinical application focusing on FNR, the XGBoost prediction model exhibited better performance among female patients, Hispanic patients, and patients without insurance coverage.

Moreover, we conducted counterfactual analysis using the XGBoost model to predict prolonged wait times among individuals of different sensitive attributes (e.g., sex, races and

ethnicities, and socioeconomic statuses). When we exchanged sex, race and ethnicity, and insurance status while keeping other features unchanged (i.e., the use of male, NHW, and patients with insurance as the privileged subgroups), improved FNRs can be observed across all sensitive attributes (Table 2).

**Group fairness evaluation.** Using measures such as disparity impact, equalized odds, and equal opportunity, we detected instances of unfairness in model predictions among patients of different sensitive attributes (e.g., sex, races and ethnicities, and patient insurance statuses). Specifically, absolute equalized odds difference can be larger than 0.1 between male and female attributes. Both equalized odds difference and equalized odds ratio were above the normal limit among patients with different races and ethnicities, and ones with/without insurance coverages, indicating the presence of unfairness among these sensitive attributes (Table 3).

Furthermore, Fig 2 illustrates the features that significantly influence the XGBoost model predictions across various sensitive attributes. Notably, patient mode of arrival and ED crowding status emerged as the most influential features for wait time predictions irrespective of these sensitive attributes. However, the impact of other features on wait time predictions varied among different sensitive attributes (Fig 2). For instance, the presence of a primary care physician had a greater influence on wait time predictions among ED visits from male patients, while abnormal vital signs at triage appeared to have a more pronounced effect among ED visits from female patients (Fig 2). Moreover, abnormal vital signs seemed to play a more substantial role in prolonged wait time predictions among ED visits of Hispanic and NHW patients compared to their counterparts, whereas having a primary care physician contributed more heavily among NHW and NHB than Hispanic patients for their prolonged ED wait time predictions (S1 Fig). Additionally, the presence of a primary care physician had a greater impact on wait time among ED visits of patients with insurance coverage, whereas speaking Spanish appeared to contribute more to prolonged wait time among uninsured patients (S2 Fig). These findings suggest that disparities exist in model predictions across different sensitive attributes.

## Discussion

The increasing application of ML prediction models in medicine has become more prevalent. However, previous studies have highlighted disparities that could introduce significant biases, potentially hindering their widespread adoption in clinical settings.[16,17] In our study, we developed an XGBoost ML prediction model, and our initial performance evaluation indicated the overall acceptable feasibility and reliability of using the model to predict prolonged wait times in the ED. However, subsequent fairness assessments revealed disparities among

**Table 3. Fairness metrics at different sensitive attributes.**

|  | male vs. female | NHW vs. NHB | NHW vs. Hispanic | Insurance Yes vs. Insurance No |
|---|---|---|---|---|
| Disparity impact difference | 0.04 | -0.02 | -0.03 | 0.01 |
| Equalized odds difference | 0.12* | 0.00 | -0.22* | 0.13* |
| Equal opportunity difference | 0.06 | 0.00 | -0.10 | 0.05 |
| Disparity impact ratio | 1.06 | 1.03 | 1.04 | 0.99 |
| Equalized odds ratio | 0.86 | 0.57* | 0.79* | 0.60* |
| Equal opportunity ratio | 1.07 | 1.00 | 1.13 | 0.94 |

Note: Group fairness was evaluated using disparity impact, equalized odds, and equal opportunities. Abbreviations: NHW, Non-Hispanic White; NHB, Non-Hispanic Black; XGBoost, eXtreme Gradient Boosting. *indicates the existence of unfairness across different sensitive attributes.

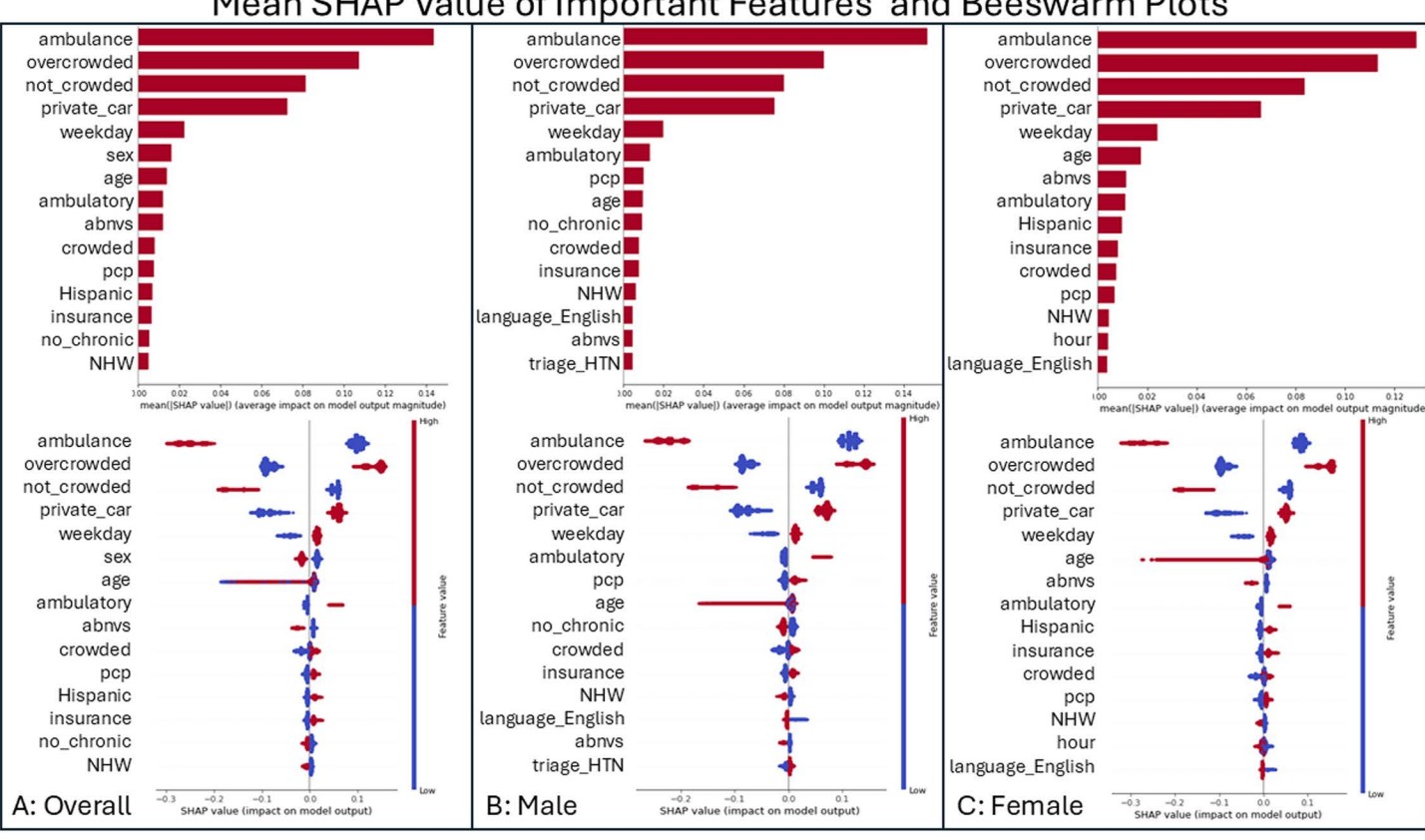

**Fig 2. Feature importance and its beeswarm plots for prolonged wait time prediction among male and female patients from XGBoost.** Fig 2 legend: Fig 2 illustrates the feature importance (upper) and beeswarm summary plots (bottom) for the predictions of the ED prolonged wait time when XGBoost prediction model is applied to male and female subgroups. This figure shows the mean SHAP value of important features (upper) and their beeswarm summary plots (bottom) when model predicts prolonged wait time among all ED visits (Panel A), ED visits from male patients (Panel B), and ED visits from female patients (Panel C).

various demographic, racial and ethnic, and insurance subgroups. While these disparities did not substantially alter the outcome predictions, their presence underscores the importance of addressing ethnic disparities and implementing rigorous quality assessments for ML model predictions as a standard practice in medical applications.

Our study not only confirms the feasibility of using ML for predicting ED patient wait times but also highlights a significant limitation in model predictions — fairness. We identify specific patient subgroups for whom the predictions are more accurate (e.g., Hispanic patients compared to NHW or NHB patients). Therefore, we advocate for a cautious approach to the use of ML models for clinical outcome prediction. Fairness evaluations should be conducted as a prerequisite before implementing final ML model applications. Our study emphasizes the importance of not only conducting general performance assessments but also ensuring the quality of model predictions. Only under such circumstances can ML models better serve our clinical practice and our patients.

In recent years, fairness evaluation has emerged as a crucial aspect in assessing the quality of ML models, aimed at identifying data biases, algorithm biases, and final outcome biases to prevent significant prediction errors.[18,19] Sex, race and ethnicity, and socioeconomic status are among the most common sources of bias during ML model development, and they were the focal points of fairness evaluation in our study.[20] Our findings revealed biases associated with these features, resulting in variations in performance accuracies.

Additionally, we emphasized the proper interpretation of the model prediction by aligning the findings to clinical applications. Our prediction model uses various features to forecast prolonged wait times. We classify these predictions using a 30-minute cutoff, based on current recommendations. This approach also benefits ED operational management. Predicting exact wait times in minutes (e.g., 15 minutes vs. 18 minutes) may not correlate well with their operational significance and cannot effectively serve as indicators to trigger intervention initiation. A categorical prediction (e.g., <30 minutes vs. ≥30 minutes) can clearly address the need for specific operational management strategies. When further interpreting the performance accuracy, we prioritized minimizing the false negative rate (FNR) due to its critical implications for patient care, where higher FNR could result in fewer predictions of patients experiencing prolonged wait times, potentially delaying their access to healthcare providers and increasing the risk of suboptimal clinical outcomes. For instance, consider a patient presenting to the ED with a chief complaint of acute abdominal pain. If this patient were experiencing acute appendicitis, and the model predicted a wait time of less than 30 minutes when the actual wait time exceeded 30 minutes (a false negative). Thus, a false negative can delay surgical consultation and antibiotic administration, which could significantly increase the risk of appendiceal perforation and sepsis, ultimately leading to a prolonged hospitalization and worse clinical outcomes. Notably, our study observed a lower FNR among Hispanic patients compared to NHW or NHB patients (24% vs. 15%, Table 3). This difference may stem from the higher rate of prolonged wait times among Hispanic patients and the higher sensitivity (recall) of the overall model predictions. Additionally, each ED visit was treated as a separate encounter for the purposes of data training and testing, rather than focusing on individual patients. This decision was made to consider patients with multiple ED visits, as their conditions during each visit can vary significantly. For instance, the same patient may present to the ED under different crowding conditions or arrive via different modes of transportation.

On the other hand, when fairness was evaluated, only the findings from equalized odds difference and equalized odds ratio showed the existence of unfairness across this sensitive attribute. This may probably be due to the calculation of these fairness metrics. The majority of current used fairness metrics tend to measure the difference among positive findings (such as true positive rate, positive predictive value), whereas our study mainly focused on the detection of FNR since the detection of FNR has far more clinical significance than any positive findings (e.g., FPR, TPR, positive prediction value). This may indicate the future adjustments of fairness metrics used in clinical applications.

This study has several strengths. We conducted a thorough quality assessment of ML models by using a well-balanced clinical data (i.e., approximately similar ED visits from patients with appropriate vs. prolonged wait times) for predicting ED prolonged wait times, including detailed fairness evaluations at both the group and individual levels. By examining common sensitive attributes such as sex, race and ethnicity, and insurance status, we ensured a comprehensive analysis of potential biases. Furthermore, we prioritized the clinical significance of our findings by focusing on the FNR as a key performance metric, rather than solely relying on overall accuracy. This approach illustrates the importance of interpretation ML model findings in clinical practice and highlights their potential impact on patient care. We believe that with such rigorous assessment and clinic-related interpretation, the integration of ML models into clinical settings can yield more meaningful outcomes.

This study is subject to several limitations that warrant consideration. Firstly, despite the overall acceptable performance of the models, disparities and unfairness were observed among various subgroups, potentially leading to inaccurate predictions. Additionally, not all potential predictors were considered in the training of the prediction model. For example, we categorized medical problems simply as no chronic disease, one chronic disease, or multimorbidity,

without specifying each chronic condition. We also did not account for nursing bias towards high ED utilizers (i.e., patients with multiple ED visits within a short period), who are often highly vulnerable populations. Their triage assessments might be biased due to their frequent ED visits and poor socioeconomic statuses. Addressing these biases and enhancing prediction accuracy will require further model modifications. Secondly, our exploration was limited to a select set of classification algorithms, and there may be other algorithms that could offer improved wait time predictions across all sensitive attributes. Our counterfactual analysis did not explicitly account for interactions between protected attributes, which could reveal intersectional effects critical for understanding disparities in wait times. Additionally, despite using time series data from 2019 to 2021, we did not address temporal autocorrelation, which could influence the reliability of results in time series data. Thirdly, as a single-centered observational study, the generalizability of our findings is restricted to EDs with similar patient populations. Fourthly, while our fairness evaluation focused on sex, race and ethnicity, and insurance status, other sensitive groups may also experience similar fairness issues.[21] Therefore, future studies should expand their exploration of ML models, incorporate a wider array of predictive features, and broaden the scope of fairness evaluations to include additional sensitive attributes. This will contribute to a more comprehensive screening process and enhance the overall quality assessment when predicting ED patient wait times.

## Methods

### Study design and setting

This is a single-centered retrospective observational study. The study hospital is an urban tertiary referral center and a level one trauma center. The hospital's ED receives approximately 120,000 visits annually. The ED has a main area staffed by ED physicians and residents who see high acuity level patients (i.e., ESI 1-3), as well as a fast-track area staffed mainly by Advanced Practice Provider who handle low acuity level patients (i.e., ESI 4-5). This study has been approved by the regional Institutional Review Board with waived informed consent (IRB#1967558-1).

### Inclusion and exclusion criteria

To predict prolonged wait times at the ED and identify features associated with prolonged wait times, we included ED visits from adult patients (≥18 years old) between January 1, 2019, and December 31, 2021, whose visits were assigned an Emergency Severity Index (ESI) level of 3 at triage. We excluded visits from patients whose wait times were not recorded, those with acuity levels other than ESI-3, or those with missing information on key sociodemographic variables (i.e., age, sex, race and ethnicity, marital status, and language proficiency) and clinical variables (i.e., insurance coverage, primary care physician assignment, comorbidities, mode of arrival at the ED, blood pressure status upon triage, and vital signs upon triage). Additionally, visits from patients who left before triage completion were excluded. The inclusion of only ESI-3 visits was due to the minimal wait times observed among visits with ESI-1 and ESI-2. Patients assigned to ESI levels 1 or 2 were typically evaluated more quickly in the ED, with their wait times generally being less than 30 minutes. As a result, the prediction model is less applicable to this group. Furthermore, our ED features a dedicated fast-track area where ESI-4 and ESI-5 patients can bypass the main ED and be seen by Advanced Practice Providers (APP) due to their lower resource requirements. Given the differences in ED patient flow management, including patients with ESI levels 4 and 5 would compromise the reliability of the prediction model. Since the primary purpose of wait time prediction is to alert patients experiencing prolonged wait times, the practical benefit of receiving wait time prediction alerts is most relevant for ED visits from patients categorized as ESI-3.

## Outcome measurement

The study outcome was patient wait time at the ED, defined as the duration from when a patient completed the triage process to when they were placed in an examination room. It was measured as a categorical variable (wait time<30minutes versus wait time≥30minutes). We selected a threshold-based approach to 1) guide decision makers, 2) enable consistent comparison across studies, and 3) assess policy goals towards reducing health disparities. Our decision to adopt a 30-minute cut off as prolonged wait time was informed by three key considerations: 1) professional consensus, 2) empirical evidence, and 3) clinical operations management. The professional consensus emerges from national surveys and recognized wait time standards. For example, A national survey of academic emergency physicians recommended that the average ED wait time should ideally be less than 30 minutes. [22] Additionally, *Emergency Physician Monthly*, a journal dedicated to emergency medicine professionals, characterized a wait time of under 30 minutes as an "excellent" standard. [23] From an empirical standpoint, the U.S. Government Accountability Office, reported that the median wait time for ESI-3 patients ranged between 15 and 60 minutes. The National Center for Health Statistics data reported a median of 30 minutes (2006 National Center for Health Statistics data) in the US. These data support the 30-minute mark as both clinically relevant and practically achievable in many ED settings. [24] Furthermore, clinical operations management can be supported by adopting a 30-minute cutoff because it has the potential to align with operational targets set by emergency medicine professionals and enables EDs to focus on interventions such as triage optimization and patient flow improvements, thus helping minimize wait times. This can result in efficient resource utilization, better patient outcomes, and satisfied patients.

## Features

The features were selected based on findings from previous studies, as well as consensus among ED administrators, including the ED flow managers, nursing team leaders, medical directors, and ED managers.[25,26] These features were grouped into four domains: sociodemographic-related, facility-related, and clinical-related. 1) Sociodemographic-related features: age, sex (male and female), marital status (single, married, and others), race and ethnicity [Non-Hispanic White (NHW), Non-Hispanic Black (NHB), Hispanic/Latino (Hispanic), Non-Hispanic Asian (NHA), and others], and language speaking (English, Spanish, and others). 2) Facility-related features: SONET score was used to determine ED crowding status (not crowded, crowded, and overly crowded),[27] and date/time of patient presented at ED, including weekend versus weekday, and clinic hours (on versus off: clinic hours were on from 8am to 5pm Monday through Friday). 3) Healthcare access -related features: insurance coverage (yes and no), primary care physician assignment (yes and no). 4) Clinical-related features: patient chronic disease condition (no chronic condition, one chronic condition, and two or more chronic conditions), patient method of arrival at ED (private car, ambulance, public transportation, and ambulatory), abnormal vital signs at triage(defined as abnormal if any one of the following criteria were met: heart rate>100 or <60, respiratory rate>20 or <12, systolic BP<90mmHg, diastolic BP<60mmHg, pulse oximetry <95%, and temperature >99.6°F or <95°F).

## Machine learning algorithms

We chose three ML algorithms to predict patient prolonged wait time including cross-validation logistic regression (CVLR), random forest (RF), and extreme gradient boosting (XGBoost). The data was divided into training (70% of the data) and testing (30% of the data) sets. The optimal performance of each ML algorithm was determined for model prediction. Among the three ML algorithms, our findings demonstrated similar accuracy performance across all models. However, the XGBoost model exhibited more balanced performance when

evaluated on both the training and testing datasets (S1 Data). Therefore, XGBoost was further evaluated for model fairness. During the derivation of the XGBoost model, hyperparameter tuning was conducted, which involved adjusting various parameters, including the number of estimators, maximum depth, gamma, subsample, and learning rate, etc. Each possible combination of these parameters was evaluated based on performance metrics such as accuracy, recall, precision, F1 score, and AUROC. GridSearchCV was employed to assess each parameter combination using five-fold cross-validation. The combination of hyperparameters that resulted in the best performance on the validation dataset was selected as the optimal configuration.

## Performance accuracy of model prediction

We assessed XGBoost model's performance based on test data set using metrics such as accuracy, true positive rate (TPR, recall, sensitivity), precision (positive predictive value), F1 score, true negative rate (TNR, specificity), false positive rate (FPR), false negative rate (FNR), and the area under the receiver operating characteristic curve (AUROC). Subsequently, we investigated the fairness of predictions across various sensitive attributes. Sensitive attributes consisted of 1) sex (male and female), 2) race and ethnicity (NHW, NHB, and Hispanic), and 3) insurance status (patients with or without insurance). In fairness evaluations, males, NHW, and those with health insurance were considered privileged groups.

## Fairness evaluation at a group level

For group-level fairness evaluation, we measured disparity impact, equalized odds, and equal opportunity. Disparity impact metrics, including disparity impact difference (the difference between the privileged and unprivileged positive prediction rates, for example, if male is the privileged group, then female will be the unprivileged group) and disparity impact ratio (the ratio of the privileged and unprivileged positive prediction rates), were reported to assess the proportion of positive predictions across groups. Equalized odds metrics, comprising equalized odds difference (the difference between TPR and FPR) and equalized odds ratio (the ratio of TPR and FPR), were used to homogenize both true and false positive rates. Equal opportunity metrics, consisting of equal opportunity difference (the difference of TPR between privileged and unprivileged groups) and equal opportunity ratio (the ratio of TPR), were reported to ensure equal identification of individuals qualified for an opportunity (i.e., sensitivity) irrespective of group differences (see detail in Table 4). Fairness was deemed achieved if the absolute

**Table 4. Detailed explanation of different fairness evaluations at a group level.**

| Fairness Evaluation | Formula | Normal Range | Interpretation |
|---|---|---|---|
| Disparate impact difference | $\text{Precision}_{priv} - \text{Precision}_{unpriv}$ | [-0.1, 0.1] | Difference of Precisions between privileged and unprivileged groups. |
| Disparate impact ratio | $\text{Precision}_{priv} / \text{Precision}_{unpriv}$ | [0.8, 1.2] | Ratio of Precision between privileged and unprivileged groups. |
| Equalized odds difference | $(\text{TPR}_{priv} - \text{TPR}_{unpriv}) + (\text{FPR}_{priv} - \text{FPR}_{unpriv})$ | [-0.1, 0.1] | Equality of true positive rates (TPRs) and False positive rates (FPRs) between the privileged and unprivileged groups. |
| Equalized odds ratio | $(\text{TPR}_{priv} - \text{TPR}_{unpriv}) / (\text{FPR}_{priv} - \text{FPR}_{unpriv})$ | [0.8, 1.2] | Ratio of TPR and FPR between privileged and unprivileged groups |
| Equal opportunity difference | $\text{Recall}_{priv} - \text{Recall}_{unpriv}$ | [-0.1, 0.1] | Absolute difference in Recalls between privileged and unprivileged groups, focusing only on the positive class. |
| Equal opportunity ratio | $\text{Recall}_{priv} / \text{Recall}_{unpriv}$ | [0.8, 1.2] | Ratio of Recall between privileged and unprivileged groups, focusing only on the positive class |

difference value was less than 0.1 and the ratios were within 0.8 to 1.2 across all measurements. The thresholds of 0.1 for differences and 0.8-1.2 for ratios are rules-of-thumb based on historical and practical perspectives. Historically, the uniform guidelines prohibit employment practices which discriminate on the grounds of race, color, religion, sex, and national origin.[28] The guidelines introduced the "80% rule" or "four-fifths rule" for detecting potential disparate impact in selection processes. Under this rule, "a selection rate for any race, sex, or ethnic group which is less than four-fifths (4/5) or (80%) of the rate for the group with the highest rate" is evidence of adverse impact. While 0.80 or (80%) rule identifies when a group is disadvantaged, 1.2 (or 120%) is often used as the upper bound to flag when a group may be receiving disproportionately favorable outcomes. This symmetric range (0.80–1.2) helps ensure fairness is evaluated symmetrically in both directions. Although originally designed for employment contexts, this threshold has been widely used in ML fairness research and incorporated in IBM's AI Fairness 360 tool, which provide well-established reference points for consistent measurement and comparison of fairness across diverse contexts.[29–33]

Additionally, SHapley Additive exPlanations (SHAP) were employed to interpret the output from the XGBoost model (global feature importance and their beeswarm summary plots to determine key features associated with prolonged wait time among different groups) to determine disparities.

### Fairness evaluation at an individual patient level

Counterfactual fairness analysis was conducted based on the definition by Kusner's et al.[34] Counterfactual fairness assumes that the ML model should not discriminate against "protected" attributes when making predictions and that the outcome prediction should remain consistent regardless of how individuals are categorized based on these attributes. For example, if sex is considered a "protected" attribute, the prediction outcome should be the same whether individual is classified as male or female. Similarly, predictions of prolonged wait time should be consistent regardless of whether the individual is classified as NHW, NHB, or Hispanic. Counterfactual fairness assessment is conducted at the individual level. [35] We reported the performance accuracy of model prediction under "counterfactual" scenarios within different subgroups (male vs. female, NHW vs. Non-White, NHB vs. non-Black, Hispanic vs. non-Hispanic, having insurance vs. no insurance).

### Data analysis

Patients were categorized into two groups based on their wait times: those with wait time of less than 30mintues and those with wait time of 30mintue or longer. A comparison of patients' sociodemographic-related, facility-related, and clinical-related features were conducted between these two groups. Continuous data were compared using the *Student-t* test for mean comparison or the *Kruskal-Wallis'* test for median comparison. Categorical data were compared using the *Chi-square* test. Two-group comparisons were performed using STATA 14.2, while XGBoost model performance assessments, and fairness evaluations were conducted using Python 3.8.

### Reporting guideline

Strengthening of the reporting of observational studies in epidemiology (STROBE) reporting guidelines were followed in this study.[36]

### Conclusion

XGBoost model demonstrated promising performance in predicting prolonged patient wait times in ED visits. However, disparities arise in predicting patients across various

demographic groups. Our model exhibited better performance among female patients, Hispanic patients, and patients without insurance coverage. To optimize the applicability of ML model predictions in clinical practice, it is essential to conduct both performance assessments and fairness evaluations.

## Supporting information

**S1 Fig. Feature importance and beeswarm summary plots for prolonged wait time prediction among patients with different races and ethnicities from XGBoost.** S1 Fig illustrates the feature importance (upper) and beeswarm summary plots (bottom) for the predictions of the prolonged wait time when XGBoost prediction model is applied to NHW (Panel A), NHB (Panel B), and Hispanic (Panel C) subgroups. Abbreviations: NHW, non-Hispanic White; NHB, non-Hispanic Black.
(TIF)

**S2 Fig. Feature importance and beeswarm summary plots for prolonged wait time prediction among patients with and without insurance coverages from XGBoost.** S2 Fig illustrates the feature importance (upper) and beeswarm summary plots (bottom) for the predictions of the prolonged wait time when XGBoost prediction model is applied to patients with (Panel A) and without insurance coverages (Panel B).
(TIF)

**S1 Data. Deidentified data for study analyses.**
(CSV)

## Author contributions

**Conceptualization:** Hao Wang, Nethra Sambamoorthi, Usha Sambamoorthi.

**Data curation:** Hao Wang, Nathan Hoot.

**Formal analysis:** Hao Wang, Nethra Sambamoorthi, Usha Sambamoorthi.

**Investigation:** Hao Wang, Nathan Hoot, David Bryant.

**Methodology:** Hao Wang, Nethra Sambamoorthi, Usha Sambamoorthi.

**Project administration:** Hao Wang, Nathan Hoot, David Bryant.

**Supervision:** Usha Sambamoorthi.

**Validation:** Hao Wang, Usha Sambamoorthi.

**Visualization:** Hao Wang, Usha Sambamoorthi.

**Writing – original draft:** Hao Wang.

**Writing – review & editing:** Hao Wang, Nethra Sambamoorthi, Nathan Hoot, David Bryant, Usha Sambamoorthi.

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
