## [Decision Letter · Decision Letter 0]

11 Dec 2024

PDIG-D-24-00431Evaluating Fairness of Machine Learning Prediction of Prolonged Wait Times in Emergency department with Interpretable eXtreme Gradient BoostingPLOS Digital Health Dear Dr. Wang, Thank you for submitting your manuscript to PLOS Digital Health. After careful consideration, we feel that it has merit but does not fully meet PLOS Digital Health's publication criteria as it currently stands. Therefore, we invite you to submit a revised version of the manuscript that addresses the points raised during the review process. Please submit your revised manuscript within 60 days Feb 09 2025 11:59PM. If you will need more time than this to complete your revisions, please reply to this message or contact the journal office at digitalhealth@plos.org. Please include the following items when submitting your revised manuscript:* A rebuttal letter that responds to each point raised by the editor and reviewer(s). You should upload this letter as a separate file labeled 'Response to Reviewers '. This file does not need to include responses to any formatting updates and technical items listed in the 'Journal Requirements' section below.* A marked-up copy of your manuscript that highlights changes made to the original version. You should upload this as a separate file labeled 'Revised Manuscript with Track Changes '.* An unmarked version of your revised paper without tracked changes. You should upload this as a separate file labeled 'Manuscript '. If you would like to make changes to your financial disclosure, competing interests statement, or data availability statement, please make these updates within the submission form at the time of resubmission. Guidelines for resubmitting your figure files are available below the reviewer comments at the end of this letter. We look forward to receiving your revised manuscript. Kind regards, Dhiya Al-Jumeily OBE, PhDSection EditorPLOS Digital Health Leo Anthony CeliEditor-in-ChiefPLOS Digital Healthorcid.org/0000-0001-6712-6626 **Additional Editor Comments (if provided):****Reviewers' Comments:**

**Comments to the Author**

Reviewer #1: This work describes the us of ML in predicting wait times for ED patients. This retrospective observational study identified unfairness across protected attributes. I have some comments/suggestions.

Is the dataset that you collected made available to other researchers?

Please try to avoid combining the AI and ML acronyms (line 30), please define individually.

Line 55 reports the number of visits, is it possible to state how many are unique patients?

There are some acronyms such as NHW and NHB which are not defined before they are used (not defined until line 308 but used much earlier).

Could you describe the distributions of characteristics such as age, gender, and race within the original dataset for example the balance of male vs female in the overall dataset. This is not clear in Table 1. E.g., perhaps add % to the middle row of Figure 1.

Table 2 provides model performance metrics. Are there any alternative methods which could be used to gain comparative results?

Line 320 provides a very brief description of the ML algorithm used (XGBoost). Can you justify the choise of XGBoost (did you test other? or does the literature support it?). Can you provide any parameter details or similar to allow reproduction. Are there any abalation studies to support this section?

The conclusion could be improved to highlight the findings of your research and its importance.

Please edit Figure 2 so that the labels are readable (e.g., increase font size). Similarly, if possible, try to increase the resolution of the image.

Reviewer #2: 1. Selection bias and sample representativeness emerge from study's focus on ESI-3 patients, excluding ESI-1, 2, 4, and 5 patients. Authors provide rationale for this choice, it limits generalizability of findings and introduces selection bias. This inclusion misses patterns in wait time disparities across acuity levels, providing picture of emergency department dynamics.

2. Study's approach to defining wait times presents limitation. Binary categorization of wait times (<30 minutes vs 30 minutes or more) is simplistic. While 30-minute cutoff stems from recommendations, threshold masks variations in wait times. Approach to analyzing wait times as continuous variable or using categories might provide insights into wait time patterns and disparities.

3. Feature engineering in study shows limitations in handling of comorbidities. Categorization into "no chronic condition," "one chronic condition," or "multiple conditions" fails to account for severity or types of conditions, which impact wait times. Categorization misses interaction effects between comorbidities and variables, overlooking relationships that influence wait times.

4. Study's model validation raises concerns. Authors mention cross-validation, they don't specify validation strategy or number of folds used, making it hard to assess robustness of methodology. There's no mention of testing for multicollinearity among predictors, which affects model's stability. Study doesn't address temporal autocorrelation, despite using time-series data from 2019 to 2021, which influences reliability of results.

5. Study's approach to fairness metrics reveals limitations. It focuses on False Negative Rate (FNR) for clinical significance, it doesn't justify why this metric matters more than others in context of emergency department wait times. Fairness thresholds (0.1 for differences and 0.8-1.2 for ratios) appear arbitrary without justification. Counterfactual analysis doesn't account for interactions between protected attributes, missing intersectionality effects that matter for understanding disparities in wait times.

Reviewer #3: The introduction section is too narrow and needs addressing broader aspects of AL/ML use in the healthcare. I would recommend to refer the following studies in healthcare recently (Line 72-73).

https://royalsocietypublishing.org/doi/full/10.1098/rsos.201823

https://www.nature.com/articles/s41598-024-51985-w

Further highlight the contributions in the Introduction section

Line 98, ‘we observed more…’. Add specific statistical outcomes/evidence rather than descriptive only.

Table 1 outcomes are important however showing no variations in the p-value. Thus all of the factors seems associated strongly.

Method: Firstly, I would like to see the detailed explanation of the dataset. Ethical considerations even if it is secondary data.

How the dataset is partitioned? Are there repetitive instances? E.g., multiple records per person?

How did you chose 30 minute to categorize the data? I would like to see support for this e.g., literature

How the categorical data and numerical data are transformed for the ML? e.g., nominal or ordinal data

Limitations of the study should be addressed

Better to explain ML algorithm in the context of this study

Reviewer #4: This is a nice example of examining a ML model for fairness across protected attribute groups. Below I include minor edits and suggestions, but include more substantial questions/concerns as well.

Major questions/concerns:

Results section, first paragraph: a lot of the text here is redundant and should be cut. The text after the mention of Table 2 (which should be Table 1) reiterates what was just said previously.

Lines 143-145: “Overall, in terms of the clinical application focusing on FNR, the XGBoost prediction model exhibited better performance among female patients, Hispanic patients, and patients without insurance coverage, which matched our findings of prolonged wait times occurring more often among patients within these subgroups (Figure 1).”

This doesn’t make sense the way you have written it. FNR is describing a type of error the model is making whereas the higher prevalence of prolonged wait times is referencing the actual data. These are not equivalent or comparable in this context. I *think* you may be trying to describe the calibration of the model but I’m not sure.

I’m wondering why only XGBoost was used and not multiple models. Training 3-6 different ML models and comparing their performance is pretty standard.

It would be more compelling to also investigate if any tweaks to the ML model(s) that might reduce or mitigate the bias found.

Lines 261 and 262: “Additionally, not all potential confounders were

considered in the training of the prediction model.” - These are not considered confounders since this is a prediction model and not a causal model. This term should be replaced with “predictors”.

Minor edits:

Throughout: gender should be called sex.

Introduction section, second paragraph: What does “overall predictions” mean here? It would be clearer to be more specific.

Introduction section, third paragraph: I would edit this sentence: “Upon identifying unfairness, appropriate adjustments can be made to the ML model predictions to enhance their accuracy and clinical applications.” to something like: “Upon identifying unfairness, appropriate adjustments made to the ML model may enhance the accuracy of the predictions and fairness in clinical applications.” I suggest this because it isn’t always possible to correct for bias (unfairness) by altering the model in some way; sometimes having more data or better quality data is the only way to improve model performance/fairness.

Results section, first sentence (line 97)- need to define NHW and NHB first time you use them in the text, e.g., Non-Hispanic White (NHW).

Results section, first paragraph minor typo (line 99): “We observed that more ED visits from females, Hispanics, and patients without insurance tended to have a prolonged wait times that those from males” - “that” should be “than”.

Results section, first paragraph (line 100): ”…and patients with

insurance, as illustrated in Figure 1.” - I’m wondering what the utility of Figure 1 is since that information is presented in Table 1. I would remove Figure 1 from the manuscript and point the reader to Table 1.

Results section, first paragraph: “General characteristics of the study patients are listed in Table 2.” - it should be Table 1

The quality of Figure 2 needs to be improved.

General comment: all of the text needs to be reviewed for grammatical errors - there are a multitude of them.

---

## [Editor Report · Decision Letter 1]

11 Jan 2025

Evaluating Fairness of Machine Learning Prediction of Prolonged Wait Times in Emergency department with Interpretable eXtreme Gradient Boosting

PDIG-D-24-00431R1

Dear Dr. Wang,

We are pleased to inform you that your manuscript 'Evaluating Fairness of Machine Learning Prediction of Prolonged Wait Times in Emergency department with Interpretable eXtreme Gradient Boosting' has been provisionally accepted for publication in PLOS Digital Health.

Best regards,

Dhiya Al-Jumeily OBE, PhD

Section Editor

PLOS Digital Health
